# Omni-Directional Capture for Multi-Drone Based on 3D-Voronoi Tessellation

Kai Cao [1,2,3], Yang-Quan Chen [1,2,3,*], Song Gao [1,2], Kun Yan [2], Jiahao Zhang [2] and Di An [3]

1 School of Mechatronic Engineering, Xi'an Technological University, Xi'an 710021, China; caokai@xatu.edu.cn (K.C.); gaos@xatu.edu.cn (S.G.)
2 School of Electronic Information Engineering, Xi'an Technological University, Xi'an 710021, China; yankun@xatu.edu.cn (K.Y.); zhangjiahao68@st.xatu.edu.cn (J.Z.)
3 School of Engineering (MESA-Lab), University of California, Merced, CA 95343, USA; dan7@ucmerced.edu
* Correspondence: ychen53@ucmerced.edu

**Abstract:** This paper addresses the multi-drone formation capture in three-dimensional (3D) environments. The omni-directional minimum volume (ODMV) 3D-Voronoi diagram algorithm is proposed for the first time to achieve the two goals of (1) forming and keeping a capture and (2) planning the control action within its safe, collision region for each drone. First, we extend the traditional 2D Voronoi diagram to the 3D environment and use the non-overlapping spatial division property of 3D Voronoi diagram to inherently avoid the collision between drones. Second, we make improvements to the problem of capture angle in our minimum area strategy and propose an omni-directional minimum volume strategy to accomplish the effective capture of a target by constraining the capture angle. Finally, the wolf pack algorithm (WPA) with variable step size is introduced to provide a movement strategy for multi-drone formations. Thus, the proposed ODMV can also achieve dynamic target and multi target capture in environments with obstacles. The Optitrack motion capture system and Crazyflie drones are used to conduct the multi-drone capture experiment. Both simulation and experimental results are included to demonstrated the effectiveness of the proposed ODMV method.

**Keywords:** multi-drone capture; 3D-voronoi; ODMV; WPA; obstacle; dynamic target

## 1. Introduction

Because an individual robot is often considered limited in its ability to handle complex tasks, group robotic systems are often used to accomplish more complex tasks. Collaborative robot capture is an important branch in robotics, which centers on how multiple robots with limited individual capabilities can coordinate to accomplish the capture or control of dangerous targets. However, when it comes to 3D environments, most existing research may have some loopholes in multi-agent containment or capture; in other words, the capture strategy may still leave the target with opportunities to escape. The theme of this paper is "omni-directional capture of targets". The advantage of multi-drone full-angle capture is the ability to fully surround static targets in all directions, such as for observation of a hazardous target and to prevent targets from escaping. Multiple robots self-organize and capture intruders in flight restricted areas, classified areas, and hazardous material storage areas, forming one or more capture circles, laying the foundation for final capture, observation, and expulsion. Existing research results are mostly focused on military [1], gaming [2], and composition [3].

There has been extensive research to address the problem of target capture [4], for example, the strategy based on reinforcement learning [5,6], game theory methods [7–9], planar reach-avoid games [10,11], consensus based [12], leader–follower [13], and others [14]. Another distinctive method is the parallel optimization method based on time minimization [15]. However, the methods noted above do not have a unified capture and

formation strategy. In addition, they do not consider obstacle avoidance among multiple robots, which leads to increased computational complexity.

Inspired by the Voronoi diagram, the formation control methods based on it have the advantages of flexibility and strong robustness, as well as inherent collision avoidance capabilities between drones [16,17]. By utilizing this approach, the Voronoi cells can be used as safe and collision-free regions for agent motion planning and collision avoidance among themselves [18–20]. These methods work well for bounded convex planes, whereas the method in [21] can be applied to unbounded convex surfaces. However, none of these approaches are suitable for environments with obstacles.

Considering obstacle avoidance, Huang et al. [22] proposed a strategy for minimizing the safety reachable region to achieve multi-pursuer capture of a single evader in a bounded simply connected domain with obstacles. Building upon [20], Zhou et al. [23] considered the case where pursuers and the evader have unequal velocities and conducted simulations in more complex non-convex environments. Wang et al. [24] based their work on [18] and considered tracking strategies for pursuers in a static obstacle environment with uncertain evader positions. Tian et al. [17] designed a buffer zone Voronoi cell (BVC) for obstacle avoidance in their work. The "move-to-centroid" strategy employed enables multiple pursuers to coordinate in capturing the evader in an obstacle environment. However, the aforementioned methods are only applicable to 2D planes and scenarios with a single target.

For the case of multi-targets, Wang et al. [25] proposed an adaptive multi-target Voronoi capture algorithm. They utilized a minimum distance criterion to enable the multi-drone formation to lock onto specific targets, repeating the process until all targets are captured. Alyssa et al. [26] investigated the minimum area capture in a multi-target scenario. In contrast to [22], Alyssa et al. imposed constraints on the drone velocities and conducted simulations in both 2D and 3D environments. However, they did not provide a clear explanation of the formation capture strategy in 3D. Although the minimum area method noted above can achieve target capture, the research on this capture strategy remains limited to the 2D plane. Moreover, the enclosing circle is not complete, allowing the targets to escape in bounded 3D environments. According to the variation of the capture angles, we found that only considering the distance factor, without constraining the capture angles, the drones will experience noticeable vibrations in order to adjust their positions during practical operations. Therefore, it is necessary to research the omni-directional capture strategy based on 3D-Voronoi in complex environments.

To situate our work in the literature, we used seven criteria to classify the type of Voronoi capture: single or multiple pursuers (S/M); single or multiple targets (S/M); 2D or 3D; collision avoidance; constraint on capture angle; capture guaranteed under the conditions in the paper; and experiments. The criteria and main related work are shown in Table 1.

**Table 1.** The criteria classification and main related work.

| Criteria/Ref. | [15] | [16] | [17] | [18] | [19] | [20] | [21] | [22] | [23] | [24] | Our Work |
|---|---|---|---|---|---|---|---|---|---|---|---|
| Captures | M | M | M | M | M | M | M | M | M | M | M |
| Targets | S | S | S | S | S | S | S | S | M | M | M |
| 2D/3D | 2D | 2D | 2D | 2D | 2D | 2D | 2D | 2D | 2D | 2D & 3D | 3D |
| Collision avoidance | Yes | No | No | No | No | Yes | Yes | Yes | No | No | Yes |
| Constraint on capture angle | No | No | No | No | No | No | No | No | No | No | Yes |
| Capture guaranteed | Yes | Yes | Yes | Yes | Yes | Yes | Yes | Yes | Yes | Yes | Yes |
| Experiments | No | Yes | No | No | No | Yes | No | No | No | Yes | Yes |

In summary, this paper utilizes the 3D-Voronoi diagram to investigate the multi-drone capture in a 3D environment. By proposing the omni-directional minimum volume

(ODMV) strategy, the drones can capture the target effectively in complex environments. Additionally, the variable step size WPA is introduced as a motion strategy. By constraining the capture angles, multi-drones can achieve omni-directional capture in a 3D environment. The structure of this paper is as follows: Section 2 introduces 3D-Voronoi and compares the effectiveness of the minimum area method and ODMV in a 2D environment. Section 3 introduces the minimum volume capture strategy and the improved ODMV strategy in 3D environments. Section 4 simulates the variable step WPA as the motion strategy and verifies the ODMV in complex and dynamic environments. Section 5 discusses the capture experiments using Crazyflie and the Optitrack positioning system. Section 6 summarizes the contributions of this paper and outlines future work.

## 2. Collaborative Capture Strategy Based on 3D-Voronoi Tessellation

### 2.1. 3D-Voronoi Tessellation

As a spatial partitioning method, the Voronoi tessellation has been studied by many scholars in recent years. In contrast to alternative formation methods, the utilization of 3D-Voronoi tessellation permits robots situated within the Voronoi cell to avoid collisions with one another, which can reduce the burden of calculation. Notably, a 3D-Voronoi diagram constitutes a continuous polygon comprising a series of vertical lines that link adjacent points in space without intersecting, thereby ensuring that each robot is assigned a distinct and non-overlapping region within the formation. The region is recognized as the 3D-Voronoi cell, and the corresponding point is identified as the generator of the 3D-Voronoi cell. Assuming that the 3D task area contains independent points and generators, the corresponding 3D-Voronoi cell for each generator can be represented as:

$$V_i^3 = \left\{ q \in \Omega^3 \mid \mathrm{sqrt}\left( \sum_{N=x,y,z} (q_n - p_{in})^2 \right) \leq \mathrm{sqrt}\left( \sum_{m=x,y,z} (q_m - p_{jm})^2 \right), i,j = 1,2,\ldots N, \forall j \neq i \right\}, \tag{1}$$

where $q$ is an arbitrary point in space, and $p_i$ is a generator. Equation (1) shows that the distance from any point within the Voronoi cell to the generator is smaller than the distance from other generators.

Unlike other formations formed by other formation methods such as leader–follower and virtual structure, the formation formed by 3D-Voronoi in this article is a flexible formation, which means that the relative positions between the generators are not fixed, and the distance between them changes as they move. Therefore, it is not necessary to expend additional computational effort on maintaining the formation, which can effectively reduce the computational burden.

### 2.2. Omni-Directional Minimum Area (ODMA) Capture Strategy

Huang et al. [20] studied the minimum area strategy in a 2D plane. Its principle is to divide a new Voronoi cell for the target based on the Voronoi tessellation and obtain the direction for each robot to minimize the area of the Voronoi cell where the target is located. Under the control of the corresponding strategy, the area of the target Voronoi cell gradually decreases until the target is captured.

In a 2D environment, if we define the Voronoi cell containing the target to be $V_k$, according to the definition of Voronoi tessellation, this Voronoi cell can be represented as:

$$V_k = \left\{ q \in V_k \mid \|q - x_k\| < \|q - x_{p_i}\|, i = 1,2,3,\ldots N \right\}, \tag{2}$$

where $N$ is the number of capture robots, and the area $A_k$ of the Voronoi cell where the target $x_k$ is mathematically defined as:

$$A_k = \int_{V_K} dq. \tag{3}$$

We calculate the negative gradient direction for this Voronoi cell, which is the fastest direction for its area reduction, that is, the direction of motion of the capture robot:

$$u_p = -\frac{\frac{\partial A_k}{\partial x_p}}{\left\| \frac{\partial A_k}{\partial x_p} \right\|} = \frac{C_i - x_p}{\left\| x_p - C_i \right\|}, \tag{4}$$

where $C_i$ is the centroid of the shared boundary $l$, and $x_p$ is the position of the capture robot. However, the minimum area strategy does not restrict the capture angle of drones, ultimately leading to the inability to capture the target. In response to this situation, this article improves on the basis of minimum area by restricting the change rate of the Voronoi cell boundary at the target. When the change rate of the final boundary is equal, it can present a better capturing effect, that is, the omni-directional minimum area strategy:

$$u_p = \frac{C_i - x_p}{\left\| x_p - C_i \right\|} \cdot \frac{\partial L}{\partial x_p}. \tag{5}$$

As shown in Figure 1, the green symbols P1–P5 represent the capture drones, and P6 is the target. The red dashed lines is the distance from the adjacent capture to the edge of the target Voronoi cell, and the blue lines is the distance from the non-adjacent capture to the target. The angles between any two robots and P6 are denoted as angle 1 to angle 5. Figure 1a is the final distribution of capture robots under the minimum area strategy. It can be observed that due to the convergence distance being satisfied between P4 and P6, the overall formation completed the capture task early. Figure 1b shows the changes of angle 1–angle 5. We can see that after the 200th iteration, significant oscillations of the angles can be observed. This is caused by the absence of constraint on the motion direction of robots. It not only fails to achieve effective capture but also affects the stable flight of drones.

Figure 2a shows the final distribution of capture robots under omni-directional minimum area strategy. It can be seen that the Voronoi cell formed by each robot and target is approximately pentagonal. From the angle changes in Figure 2b, the angles tend to be equal, specifically converging to 72 degrees. This indicates that the ODMA is effective in achieving a successful capture of target.

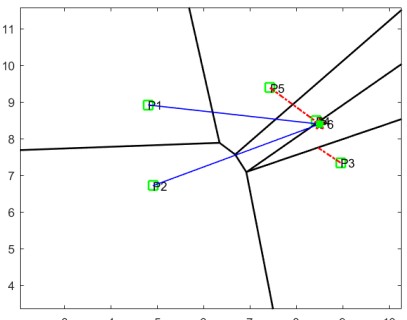

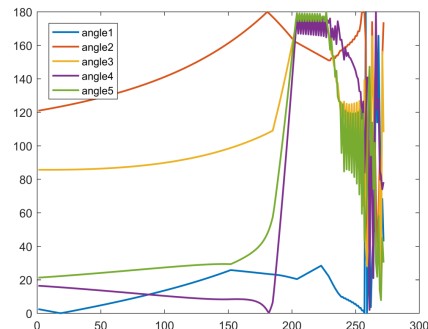

**(a)** The final distribution of drones and target   **(b)** The variability of the angle between two robots

**Figure 1.** The capture result of minimum area strategy.

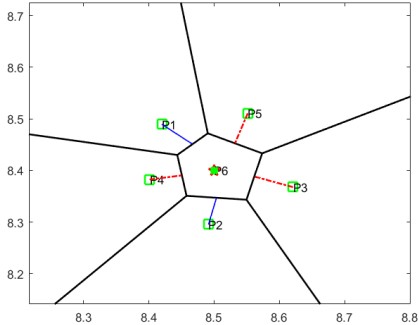

(**a**) The final distribution of drones and target

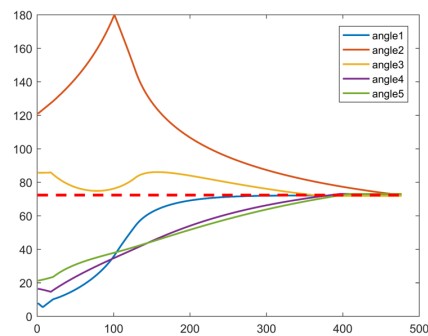

(**b**) The variability of the angle between two robots

**Figure 2.** The capture result of omni-directional minimum area strategy.

### 2.3. Variable Step Wolf Pack Algorithm (WPA)

After determining the capture direction of ODMA, we introduce the variable step WPA as the motion method of drones. The details of WPA can be found in [27], and the fixed step of traditional WPA is improved as a variable step:

$$step = \frac{d_{max}}{d_{min}} |x - X_g|, \tag{6}$$

where $|x - X_g|$ is the distance between drones and target, and $d_{max}$ and $d_{min}$ represent the maximum and minimum distance between drones and target in the capture process. The ratio of these distances is used to prevent the drones from having excessively large or small steps. As the drones approach the target, the ratio will gradually approach 1, enabling a smooth transition of step.

### 2.4. Improved Artificial Potential Field

During the capture process, drones need to avoid the obstacle in their path, therefore, an improved artificial potential field (IAPF) is adopted as the collision avoidance strategy for the drones. Modifications have been made to address the issues of traditional APF, which can be found in our previous work [28].

The improvement in the target reachability involves two steps. First, a limit is set for the effective range of the repulsive potential field. Second, a distance factor is introduced into the calculation of the repulsive force, further constraining the repulsive force based on the distance to the target. The improved calculation for the repulsive potential field is as follows:

$$U_{rep}(x) = \begin{cases} \frac{\varepsilon}{2}(\frac{1}{d_m} - \frac{1}{d_0})^2 d^n, d(x, X_o) < d_m \\ \frac{\varepsilon}{2}(\frac{1}{d(x,X_o)} - \frac{1}{d_0})^2 d^n, d_o < d(x, X_o) < d_m. \end{cases} \tag{7}$$

In the improvement of the local optima, a criterion factor is introduced as an indicator to determine whether a drone is trapped in a local optima.

$$\begin{cases} |F_{att} + F_{rep}| \leq 0 + \varepsilon \\ |\angle F_{rep} - \angle F_{att}| = \pi + \delta, \end{cases} \tag{8}$$

where $\varepsilon$ and $\delta$ are infinitesimal values of resultant force and angle. When it is determined that a drone is trapped in a local optima, the drone's movement follows the following rules:

$$x_{new} = x_{old} + t_s \sin\frac{2\pi(i-1)}{n}, (i = 1, 2, \ldots n), \tag{9}$$

where $x_{new}$ is the newly generated target, and $x_{old}$ is the drone position after falling into local optima. By improving the artificial potential field, the target reachability and local optima in traditional artificial potential field methods will be solved.

## 3. 3D-Voronoi Capture Strategy with ODMV

### 3.1. Minimum Volume Capture Strategy

Based on the research in [20], this section applies the minimum area capture strategy to the multi-drone formation capture in 3D space. Due to the difference between 3D and 2D space, the minimum area capture strategy becomes the minimum volume capture strategy in 3D space. The Voronoi cell where the target is located can be represented as:

$$V_k^3 = \left\{ q \in \Omega^3 | sqrt \left( \sum_{N=x,y,z} (q_i - x_k)^2 \le \sum_{m=x,y,z} (q_j - x_p)^2 \right), i,j = 1,2,\dots N, \forall j \ne i \right\}. \tag{10}$$

The volume of $V_k$ is:

$$V_{V_k} = \int_{V_k} dq, q \in V_k. \tag{11}$$

Taking its derivative and combining Equation (10), the change rate of its volume can be expressed as:

$$\dot{V}_{V_k} = \sum_{j \in N_k} \int_{A_j} \left[ \frac{(x_p - q) \dot{x}_p}{\|x_p - x_k\|} - \frac{(x_k - q) \dot{x}_k}{\|x_p - x_k\|} \right] dq, \tag{12}$$

where $N_k$ is the surface number of $V_k$, $A_j$ is the $j$th face of $V_k$, and $x_k$ and $x_p$ are the coordinates of the target and the capture drones, with common factors:

$$\dot{V}_{V_k} = \sum_{j=1} \int_{A_j} dq \left[ \frac{\left( x_p - \int_{A_j} q dq / \int_{A_j} dq \right)}{\|x_p - x_k\|} \right] - \sum_{j=1} \int_{A_j} dq \left[ \frac{\left( x_k - \int_{A_j} q dq / \int_{A_j} dq \right)}{\|x_p - x_k\|} \right]. \tag{13}$$

According to the physical meaning of integration, the common factor in Equation (13) can be expressed as:

$$A_k = \int_{A_j} dq, C_k = \int_{A_j} q dq / \int_{A_j} dq, \tag{14}$$

where $A_k$ is the area of the $j$th surface , and $C_k$ is the centroid of $A_j$. Then Equation (13) can be simplified as:

$$\dot{V}_{V_k} = \sum_{j=1} A_k \left[ \frac{(x_p - C_k)}{\|x_p - x_k\|} \right] \dot{x}_p - \sum_{j=1} A_k \left[ \frac{(x_k - C_k)}{\|x_p - x_k\|} \right] \dot{x}_k. \tag{15}$$

During the process of multi-drones approaching the target, the change of the target 3D-Voronoi is directly related to the drones adjacent to the target, so the change rate of the Voronoi cell can also be expressed as:

$$\dot{V}_{V_k} = \frac{\partial V_k}{\partial x_k} \dot{x}_k + \sum_{j=1}^{N_n} \frac{\partial V_k}{\partial x_{p_j}} \dot{x}_{p_j} + \sum_{i=1}^{N-N_n} \frac{\partial V_k}{\partial x_{p_i}} \dot{x}_{p_i}, \tag{16}$$

where $N_n$ is the number of the 3D-Voronoi cells adjacent to $x_k$. Combined with Equation (15), it can be obtained:

$$\sum_{j=1}^{N_n} \frac{\partial V_k}{\partial x_{p_j}} \dot{x}_{p_j} + \sum_{i=1}^{N-N_n} \frac{\partial V_k}{\partial x_{p_i}} \dot{x}_{p_i} = A_k \sum_{j=1}^{N_k} \frac{\left( x_{p_j} - C_k \right)}{\left\| x_{p_j} - C_k \right\|}. \tag{17}$$

In our research, the movement of non-adjacent drones is not directly related to the changes in the target 3D-Voronoi cell, so Equation (17) can be written as:

$$\sum_{j=1}^{N_n} \frac{\partial V_k}{\partial x_{p_j}} \dot{x}_{p_j} = A_k \sum_{j=1}^{N_k} \frac{\left(x_{p_j} - C_k\right)}{\left\|x_{p_j} - C_k\right\|}. \tag{18}$$

Using the negative gradient direction with the fastest volume reduction, the motion direction of the adjacent drone $x_p$ can be obtained:

$$u_p = -\frac{\partial V_k}{\partial x_p} \Big/ \left\|\frac{\partial V_k}{\partial x_p}\right\| = \frac{C_i - x_p}{\left\|x_p - C_i\right\|}, \tag{19}$$

where $C_i$ is the centroid of the surface adjacent to the target. After combining the minimum volume capture strategy with 3D-Voronoi tessellation, the motion direction of adjacent robots can be obtained.

However, because the motion of adjacent robots are not limited, the minimum volume strategy cannot guarantee the capture of the target, which means that in some dynamic scenarios, it will increase the space for the escape of dynamic target. Therefore, improvement of the minimum volume method is needed.

### 3.2. Omni-Directional Minimum Volume (ODMV)

The ODMV strategy refers to constraint of the motion directions of the drones during the capture process. This allows the drone formation to achieve an omni-directional capture of the target, resembling a regular polygon. The visual representation of the capture is depicted in Figure 3. The green stars is the target.

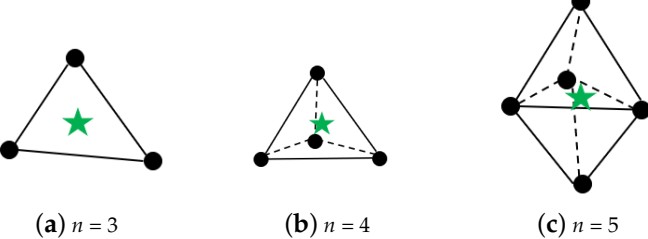

(**a**) $n = 3$　　　　　(**b**) $n = 4$　　　　　(**c**) $n = 5$

**Figure 3.** Visual representation of omni-directional capture of target under a differing number of drones.

**Theorem 1.** *To capture the target in all directions by the minimum volume strategy, the same change rate on each surface of the target 3D-Voronoi cell must be maintained.*

**Proof.** According to the nature of regular polygon, each surface is equal:

$$A_1 = A_2. \tag{20}$$

Derivation of Equation (20):

$$\frac{\partial A_1}{\partial x_{p_1}} = \frac{\partial A_2}{\partial x_{p_2}}, \tag{21}$$

which means the motion rate of each surface is the same. By the volume calculation of the Voronoi cell, it can be obtained that under the minimum volume strategy, the volume change of the target 3D-Voronoi cell is:

$$\dot{V}_{V_k} = \frac{A}{x_p - x_k}\left[-(x_p - C_i) - (x_k - C_i)\,\dot{x}_k\right]. \tag{22}$$

Because the volume change of the target 3D-Voronoi cell is directly related to adjacent drones, Equation (22) can be changed to:

$$\dot{V}_{V_k}\bigg|_{p_i, i=1,2,\dots N} = \frac{A}{x_p - x_k}\left[-(x_p - C_i) - (x_k - C_i)\,\dot{x}_k\right]. \tag{23}$$

The second derivative of Equation (23) is:

$$\ddot{V}_{V_k}\bigg|_{p_i, i=1,2,\dots,N} = \frac{A}{x_p - x_k}\left[-\dot{x}_{p_i} - \dot{x}_k^2 - (x_k - C_i)\,\ddot{x}_k\right]; \tag{24}$$

when $i$ takes different values, the corresponding results will not be the same. Thus, if you want to have the same rate of change of the volume in each direction, it is necessary to ensure that:

$$\dot{x}_{p_1} = \dot{x}_{p_2} \Rightarrow \frac{\partial A_{p_1}}{\partial x_{p_1}} = \frac{\partial A_{p_2}}{\partial x_{p_2}}. \tag{25}$$

The motion speed of the drone adjacent to the target tends to be equal, consistent with the conclusion drawn by Equation (5). Therefore, the motion rate of adjacent drones becomes:

$$u_p = -\frac{\partial A}{\partial x_p} \cdot \frac{\partial V_k}{\partial x_p} \bigg/ \left\|\frac{\partial V_k}{\partial x_p}\right\| = \frac{\partial A}{\partial x_p} \cdot \frac{C_i - x_p}{|x_p - C_i|}. \tag{26}$$

$\square$

**Theorem 2.** *To ensure that drones achieve uniform capture in terms of angles, it is necessary to maintain equal distance between the drones and the target.*

**Proof.** Let the angle between any two adjacent drones $p_1$ and $p_2$ be denoted as $\theta$. Then the angle can be expressed using the cosine rule as follows:

$$\theta = \arccos \frac{L_1^2 + L_2^2 - L_3^2}{2L_1L_2}, \tag{27}$$

where $L_1$ and $L_2$ represent the distance between $p_1$ $p_2$ and the target, and $L_3$ is the distance between two adjacent drones. In achieving the omni-directional capture of the target, the distance between each drone and the target is the same, denoted as $L_1 = L_2$; the equation can be simplified as follows:

$$\theta = \arccos \frac{2L_1^2 - L_3^2}{2L_1L_2}. \tag{28}$$

The derivative of the angle $\theta$ with respect to time is given by:

$$\frac{d\theta}{dt} = \frac{-1}{\sqrt{1 - X^2}} \frac{\partial X}{\partial t}, \tag{29}$$

where $X$ can be denoted by:

$$X = \frac{2L_1^2 - L_3^2}{2L_1}, \tag{30}$$

$$\frac{\partial X}{\partial t} = \frac{L_1 L_3 \frac{\partial L_3}{\partial t} - L_3^2 \frac{\partial L_1}{\partial t}}{L_3^3}. \tag{31}$$

Based on Equation (31), it can be seen that when the distance between adjacent drones and the target approach $L_1 \rightarrow L_3$, the angle variation between any two adjacent drones tends to be the same, which means that $\frac{\partial X}{\partial t} = 0$.

Therefore, the drones are capable of achieving the omni-directional capture of the target, with the angle between any two drones and the target tending to be equal. $\square$

### 3.3. Convergence Proof

3.3.1. Convergence Proof of 3D-Voronoi Process

In the 3D environment of this article, let the cost function of the system be:

$$E_V(p) = \int_{V_i} (x - x_p)\rho(x)dx, i = 1, 2, 3 \ldots, n. \tag{32}$$

The cost function is discretized to obtain:

$$E_V(p) = \sum_{V_j} \sum_{j=1} (x - x_p)\rho(x), \tag{33}$$

where $x \in Q^3$ is an independent point, $x_p$ in the position of drone, and $\rho(x)$ is a priority function: the higher value of this function represents the higher the priority. The cost function estimates the error during the drone movement through the priority function within the region. The proof of minimizing the cost function by the 3D-Voronoi tessellation is as follows:

Set the position change of the drone to $\varepsilon$, then the cost function is:

$$E_V(x_p) - E_V(x_p + \varepsilon) = \int_{V_i} \rho(x)\varepsilon[\varepsilon + 2(x_p - x)]dx. \tag{34}$$

Simplify the above formula to:

$$\frac{E_V(x_p) - E_V(x_p + \varepsilon)}{\varepsilon} = \int_{V_i} \rho(x)[\varepsilon + 2(x_p - x)]dx. \tag{35}$$

Make the position change $\varepsilon$ tend to 0. We can conclude Equation (35) by L'Hopital's rule:

$$\dot{E}_V(x_p) = 2\int_{V_1} \rho(x)(x_p - x)dx. \tag{36}$$

Make the derivative part 0, then we obtain:

$$x_p = \frac{\int_V x\rho(x)dx}{\int_V \rho(x)dx}. \tag{37}$$

The position of generator is the centroid of Voronoi cell, which means that 3D-Voronoi tessellation can minimize the cost function of the system.

3.3.2. Convergence Proof of Minimum Volume Strategy

To prove that the target 3D-Voronoi cell no longer increases under ODMV, Equation (19) is substituted into Equation (22), as follows:

$$\dot{V}_{V_k} = \frac{A}{x_p - x_k}\left[(x_p - C_i)\,\dot{x}_p - (x_k - C_i)\,\dot{x}_k\right]. \tag{38}$$

According to Equation (26), it can be simplified as follows:

$$\dot{V}_{V_k} = \frac{A}{x_p - x_k}\left[-\frac{\partial A}{\partial x_p}(x_p - C_i) - (x_k - C_i)\,\dot{x}_k\right]. \tag{39}$$

To make $\dot{V}_{V_k} = 0$:

$$-\frac{\partial A}{\partial x_p}(x_p - C_i) - (x_k - C_i)\,\dot{x}_k = 0, \tag{40}$$

$$\dot{x}_k = \frac{(C_i - x_p)}{C_i - x_k} \cdot \frac{\partial A}{\partial x_p}. \tag{41}$$

We can see that the value of $\dot{V}_{V_k}$ is always no greater than 0, and only when the target moves towards the centroid of the shared surface will it be taken as 0. Therefore, under the influence of the motion direction $u_p$, the volume of the target 3D-Voronoi will never increase.

To prove that the distance between the drone and the target is always reduced when $\dot{V}_{V_k} = 0$ , define $D$ as:

$$D = x_p^2 - x_k^2. \tag{42}$$

The derivative of $D$ is:

$$\dot{D} = 2(x_p - x_k)(\dot{x}_p - \dot{x}_k). \tag{43}$$

Substituting Equations (41) and (43):

$$\dot{D} = 2(x_p - x_k)\left(\frac{(C_i - x_p) - (C_i - x_k)}{(C_i - x_k)(C_i - x_p)}\right)\frac{\partial A}{\partial x_p}. \tag{44}$$

Due to the nature of 3D-Voronoi tessellation, the distance from $C_i$ to the target is equal to the distance from the capture drone, $C_i - x_k = C_i - x_p$, so Equation (44) can be written as:

$$\dot{D} = -2\frac{(x_k - x_p)^2}{(C_i - x_p)^2}\frac{\partial A}{\partial x_p}, \tag{45}$$

which represents that the distance between the drone and the target is always decreasing. Figure 4 represents the overall logical framework of the algorithm. The pseudocode is shown in Algorithm 1.

---

**Algorithm 1** The pseudocode of the ODMV algorithm.

---

1 *Initialization*
2　　Initialize drones' position $\{x_p\}_{i=1}^{n}$ ;number $n$; task area $Q \in R^3$ ; target $\{x_e\}$
3 *Procedure*
4　　Generate 3D-Voronoi cells $\{V_i^3\}_{i=1}^{n+1}$ and calculate the centroids $\{z_i\}_{i=1}^{n+1}$
5　　Ensure the position of the head-wolf
6 *while* Drones and target is in capture distance && Capture angle is reach the standard
7　　Drones move to the head-wolf drone.
8　 *if* The current 3D-Voronoi cell is adjacent to the target 3D-Voronoi cell
9　　Calculate the centroid of the surface adjacent to the target point $Z_i$
10　　Adjacent robot to centroid $Z_i$
11　　Calculate the change rate for each surface and make corrections to drones
8　　*if* The capture angle of drones satisfy the omni-directional capture
9　　　The motion directional provided by the ODMV is the moving direction of drones.
10　　*else*
11　　　A constraint is imposed of the motion direction based on the variation of the capture angles, resulting a new motion direction.
12　　*end if*;
12　*else*
13　　Move directly to the target
14　*end if*
15 *end while*

---

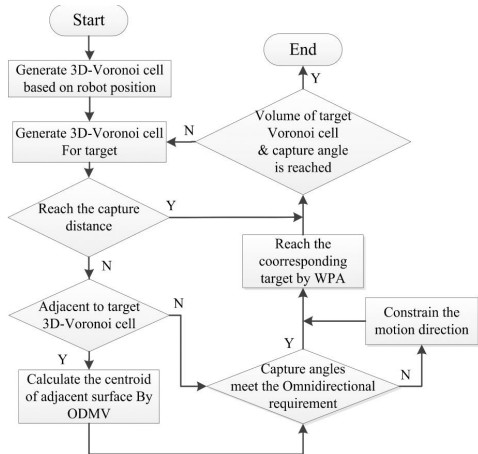

**Figure 4.** The Overall logical framework.

## 4. Simulations Analysis and Verification

This section simulates the algorithm proposed in the article. The simulation platform is Matlab 2016b under the Windows system. The simulation environment is a 3D space with a size of $10 \times 10 \times 10$ m. The positions of the drone and the target are represented by black and green squares, respectively. The red line represents the movement trajectory of the capture. To minimize the volume of the target 3D-Voronoi cell, we set the interception distance to 0.1 m. The capture angle of drones to target is used as the effectiveness indicator of the process. In Section 4.1, the capture based on WPA and minimum volume strategy are shown as the comparison process, whereas in Sections 4.2–4.4, we omitted the capture process and use the data of these two processed as the comparison. In Sections 4.2–4.4, the simulations of dynamic targets, obstacle, and multiple targets environments, respectively, are described.

### 4.1. Formation Capture under Non-Obstacle Environment

According to the size of the 3D space, we set the step-size parameter of WPA as $d_{max} = 9$, $d_{min} = 0.5$. Figure 5 shows the capture based on the variable step-size WPA cannot guarantee the capture when the target is located on the side of the formation. The reason for this problem is that the head-wolf drone is far from the target.

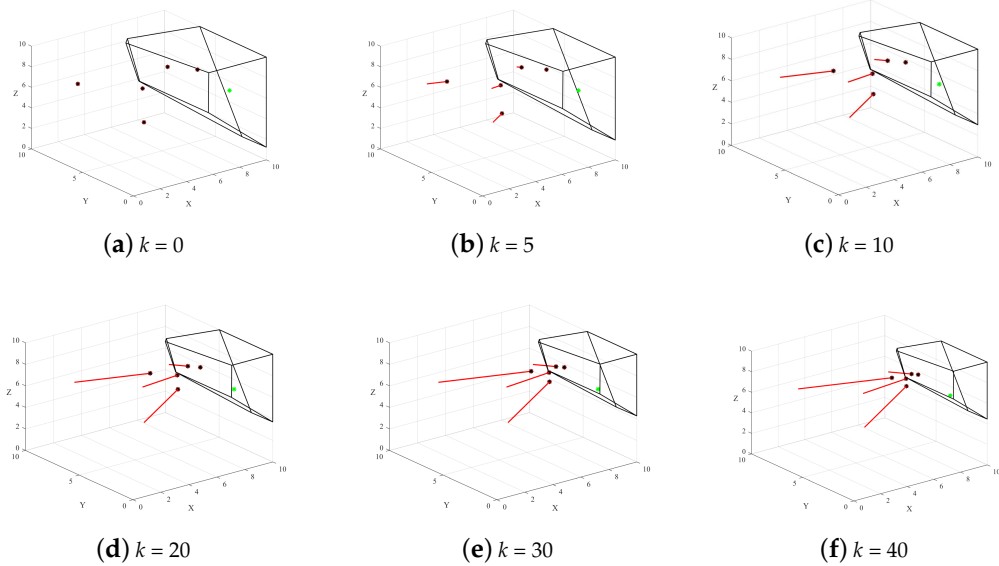

| (**a**) $k = 0$ | (**b**) $k = 5$ | (**c**) $k = 10$ |
| (**d**) $k = 20$ | (**e**) $k = 30$ | (**f**) $k = 40$ |

**Figure 5.** 3D-Voronoi capture without minimum volume strategy.

Figure 6 depicts the capture process under the minimum volume. The final distribution is shown in Figure 6f. Although the minimum volume strategy addresses the capture failure

of WPA, the capture process ends prematurely due to the distance between one of the drones and the target reaching the capture distance. As a result, the volume of the target 3D-Voronoi cell cannot be minimized.

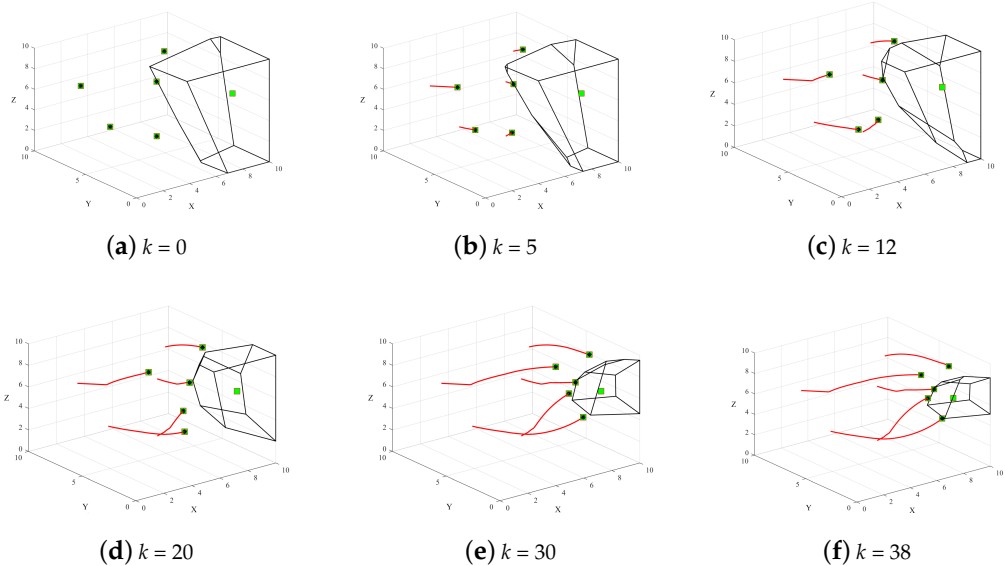

**Figure 6.** 3D-Voronoi capture based on minimum volume strategy.

Through the comparison process shown in Figures 6 and 7, we can see that after making improvements to the minimum volume strategy, the drone's motion can be effectively constrained, allowing more drones to participate in the capture process, which enables the formation to effectively capture the target.

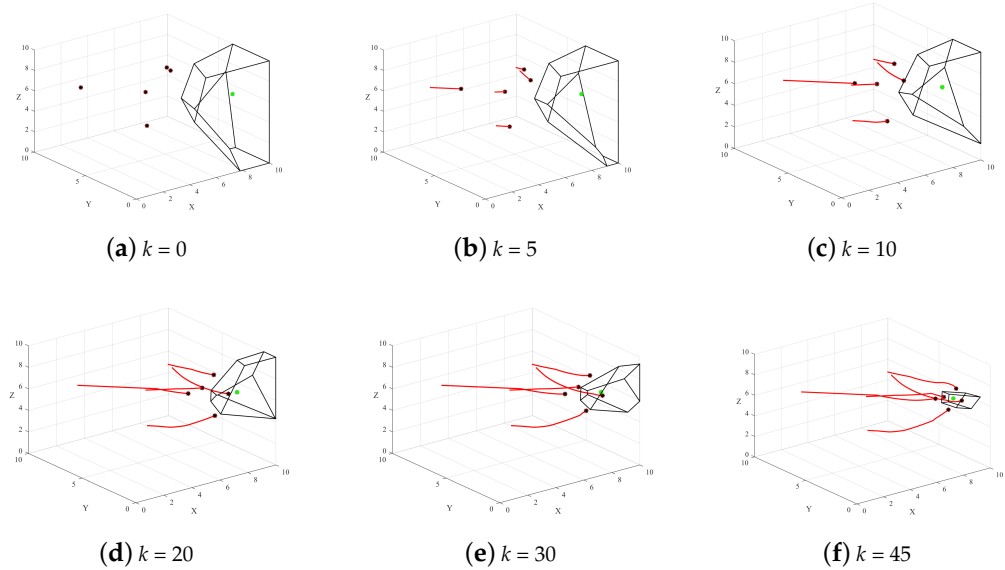

**Figure 7.** 3D-Voronoi capture based on ODMV strategy.

Figure 8 is a comparison of the three processes. Red and green curves are the volume changes before and after adding the minimum volume strategy, and the blue curve indicates the volume change after adding the ODMV strategy.

As can be seen from Figure 8, WPA cannot minimize the volume of the target 3D-Voronoi cell, and it is not possible to form a good capture for the target. The green curve shows that the target has finally been captured, but the volume cannot be minimized.

According to the changes in the blue curve, the ODMV can minimize the volume of the target 3D-Voronoi cell.

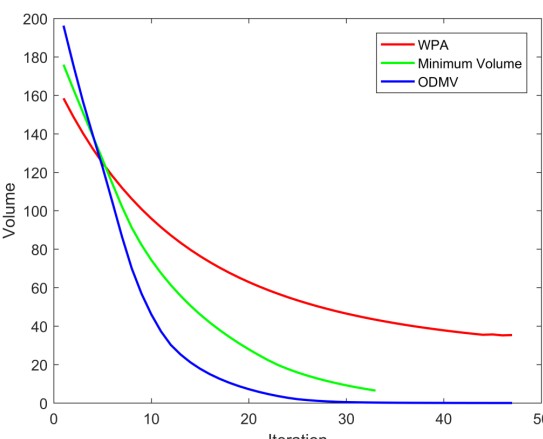

**Figure 8.** Volume change before and after adding minimum volume strategy.

To visually compare the capture angle during the capture process, the capture angle is introduced here:

$$\delta = \frac{\alpha}{2\pi}, \tag{46}$$

where the capture angle is $\alpha$. The changes in the capture angle of the drone formation to the target are shown in Figure 9.

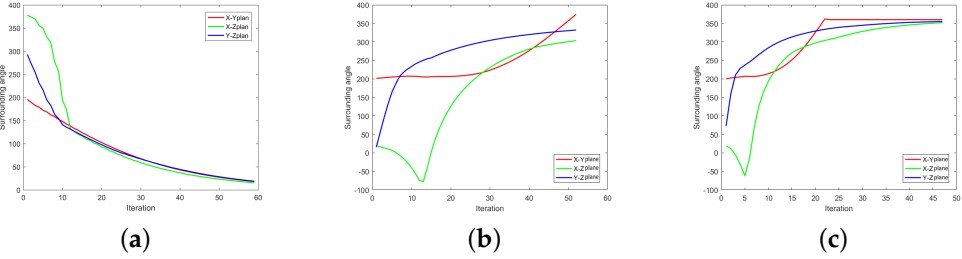

(a)              (b)              (c)

**Figure 9.** Capture angle change before and after adding minimum volume strategy. (**a**) Variable step size WPA formation capture; (**b**) Minimum volume formation capture; (**c**) ODMV formation capture.

Figure 9 shows the change in the capture angle of the multi-drone formation. The capture angles of the $X-Y$, $X-Z$, and $Y-Z$ planes are used to reflect the encirclement of the formation against the target in a 3D environment.

### 4.2. Formation Capture under Dynamic Target Environment

Under the dynamic target environment, the motion trajectory of the target is represented by a blue curve, and the robot's motion trajectory is represented by a red curve. We can see that under the ODMV strategy, multiple drones are eventually able to complete the omni-directional capture of this dynamic target.

As can be seen in Figure 10, the ODMV can still complete the capture of this target under the case of a dynamic target. Similar to Section 4.1, we conducted a comparative experiment on the capture angles in WPA, minimum volume, and the ODMV, as shown in Figure 11.

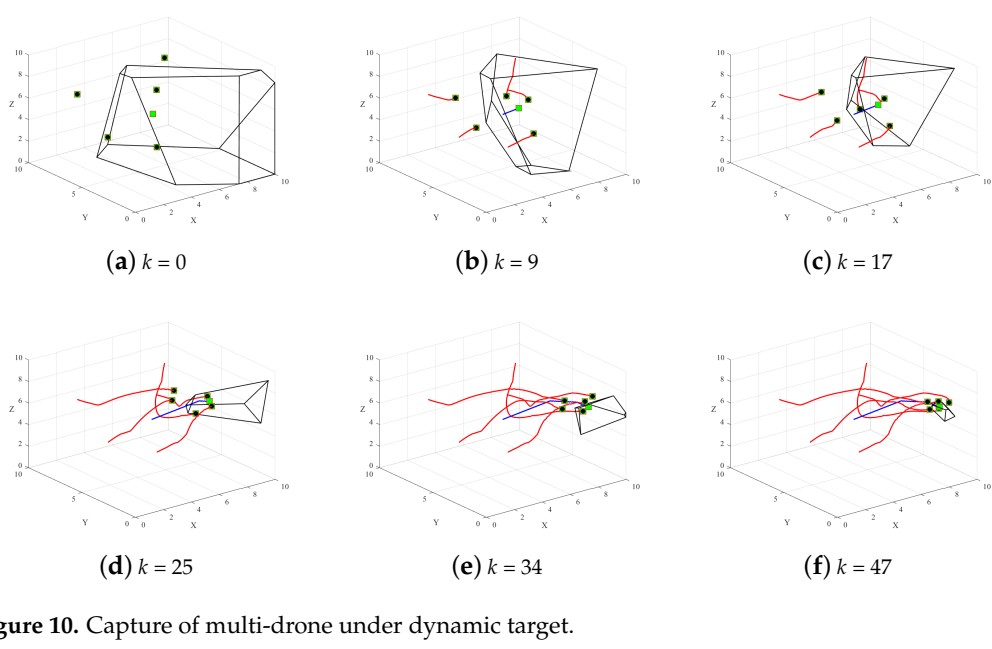

**Figure 10.** Capture of multi-drone under dynamic target.

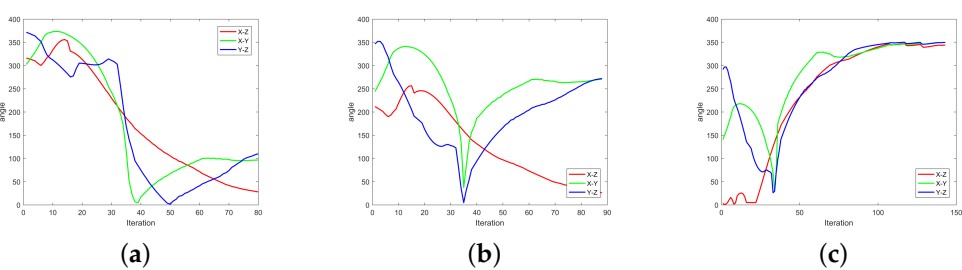

**Figure 11.** The change of capture angles under dynamic target. (**a**) Variable step size WPA formation capture; (**b**) Minimum Volume formation capture; (**c**) ODMV formation capture.

Figure 11 illustrates the variation of capture angles under a dynamic target. From Figure 11a,b, it can be observed that both the WPA and the minimum volume strategy fail to achieve effective encirclement of the target. In Figure 11c, the capture angles finally reach to 360, and the omni-directional capture is accomplished.

### 4.3. Formation Capture under Obstacle Environment

In the obstacle environment, the process of formation capture is shown in Figure 12. In order to facilitate the observation of the capture process, we adjusted the perspective of the simulation picture and the transparency of the surface of the obstacle:

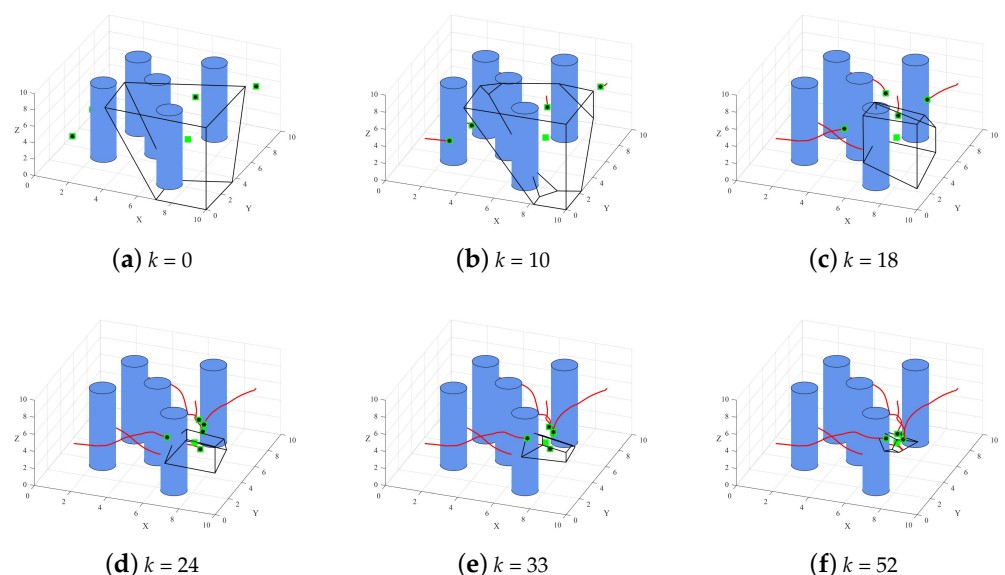

**Figure 12.** Capture of multi-drone under obstacles.

In the simulation process of Figure 12, we use the artificial potential field (APF) method for obstacle avoidance between drones and obstacles; an overview can be found in our previous work [26]. The variation of capture angles are shown in Figure 13.

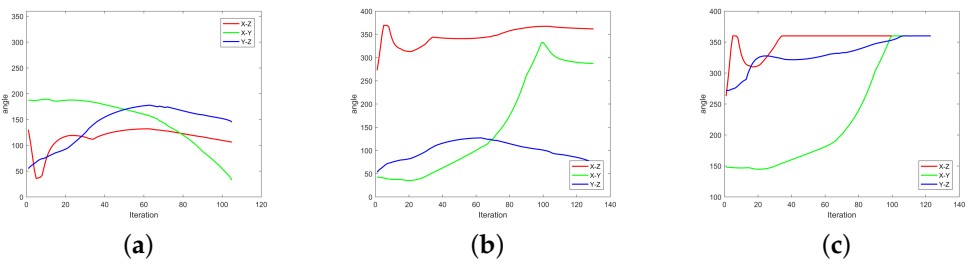

**Figure 13.** The change of capture angles under obstacles. (**a**) Variable step size WPA formation capture; (**b**) Minimum Volume formation capture; (**c**) ODMV formation capture.

Figure 13 shows that the ODMV strategy can achieve the omni-directional capture of the target, with capture angles in each plan reaching 360 degrees. On the other hand, in the capture process using the WPA and the minimum volume strategy, the capture process of the drone formation ends prematurely due to the lack of constraints on the motion direction. As a result, they fail to achieve effective omni-directional capture of the target.

### 4.4. Formation Capture under Multiple Targets

The ODMV is also appropriate for multi-target capture. In many practical scenarios, the drone formation needs to capture more than one target. Therefore, in this section, simulations are conducted with two targets, and the number of capture drones is increased from 5 to 10. The drone formation determines the capture target based on the distance. The capture process is shown in Figure 14.

Figure 14 shows that the capture strategy based on the ODMV can accomplish the capture process in a two target mission. This conclusion will also be suitable for three or more targets.

Figure 15 shows the capture angle variation of the target in the upper left corner during the capture process. The final angles in Figure 15a and Figure 15b do not satisfy the effective capture. In Figure 15b, due to the absence of constraints on the movement

direction, significant oscillations occur in the X-Y plane. Figure 15c shows the ODMV achieves the omni-directional capture.

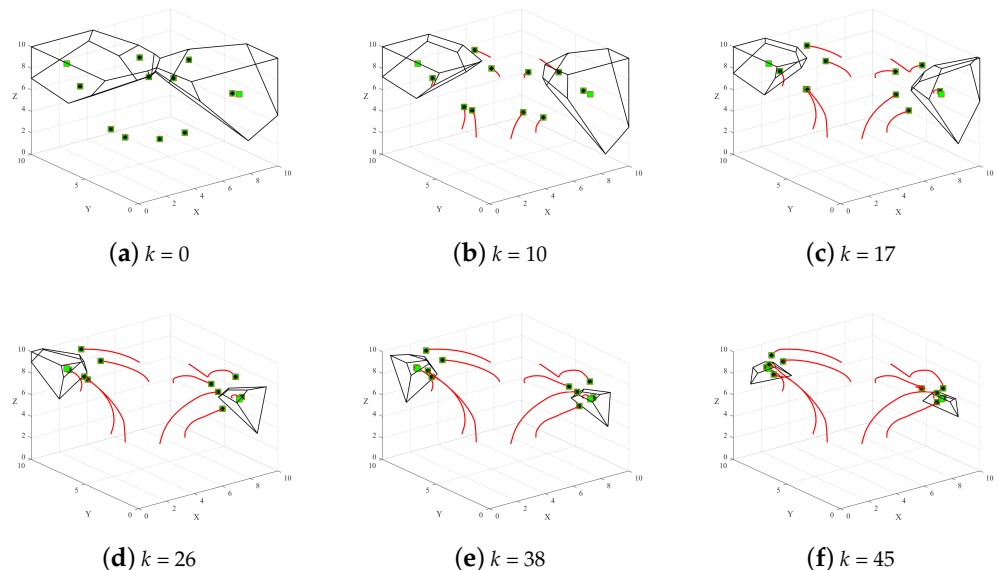

(**a**) $k = 0$　　　　　　(**b**) $k = 10$　　　　　　(**c**) $k = 17$

(**d**) $k = 26$　　　　　　(**e**) $k = 38$　　　　　　(**f**) $k = 45$

**Figure 14.** Capture of multi-drones under two targets.

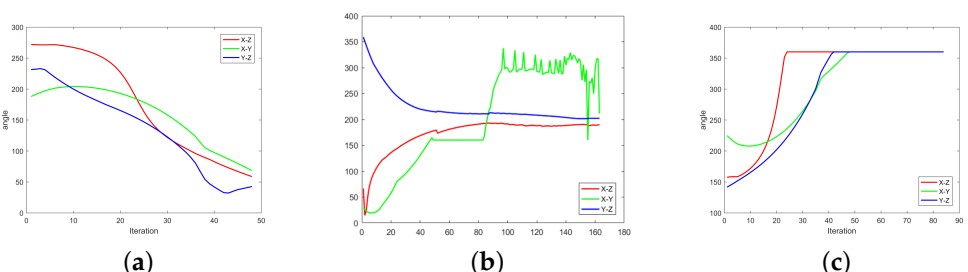

(**a**)　　　　　　　　(**b**)　　　　　　　　(**c**)

**Figure 15.** The change of capture angles under two targets. (**a**) Variable step size WPA formation capture; (**b**) Minimum Volume formation capture; (**c**) ODMV formation capture.

Table 2 presents a comparison between the average distance and the capture angle between the drone and the target after the final convergence. The capture angle here refers to the angle of encirclement formed by the drone formation and the target when the iteration is finally completed.

Through the data comparison in Table 2, we can see that during the capture process after adding the ODMV, the average distance and capture angle of the drone formation to the target are improved, and this result also applies to the complex environment of dynamic targets and obstacles.

**Table 2.** Comparison before and after adding minimum volume strategy.

| Strategies (Section)/Indicators | Distance between Drones and Target (m) | Capture Level |
|---|---|---|
| Variable step WPA formation capture (Section 4.1) | 2.799 | 12.50% |
| Minimum volume formation capture (Section 4.1) | 0.788 | 88.42% |
| ODMV formation capture (Section 4.1) | 0.302 | 99.26% |
| Formation capture under dynamic target (Section 4.2) | 0.201 | 99.05% |
| Formation capture under obstacle environment (Section 4.3) | 0.205 | 98.63% |
| Formation capture under multiple targets (Section 4.4) | 0.310 | 98.26% |

## 5. Crazyflies Capture Experiment

We set an indoor experiment environment with size of 5 m × 5 m × 3 m, as shown in Figure 16a. The red circles are labeled Cemara, Crazyflies and Reflective sphere separately. Eight infrared cameras are evenly installed above the area, which can be used to track up to six different targets. For the capture experiment, we selected four Crazyflies, as shown in Figure 16b. Each Crazyflie has a wheelbase of 92 mm and weighs approximately 30 g, making it suitable for experiments in small indoor spaces. In Figure 16a, the Optitrack motion capture cameras are shown along with the four Crazyflie drones. We attached three or four reflective spheres to each Crazyflie, which can reflect the infrared light emitted by the Optitrack cameras, facilitating the motion capture system to acquire the position of drones.

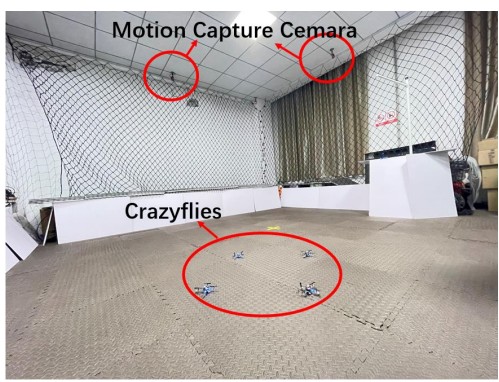
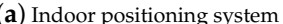
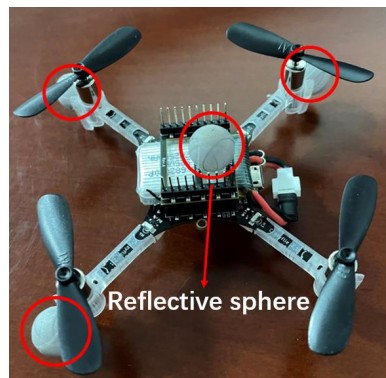

(**a**) Indoor positioning system　　　(**b**) Crazyflie with four reflective spheres

**Figure 16.** Indoor experimental platform.

The structure of the drone control system is shown in Figure 17. The Optitrack motion capture system is utilized to track the real-time positions of the drones. The captured positions are then transmitted to the ground control station (GCS) through the robot operating system (ROS). The GCS employs the ODMV algorithm to calculate the target positions for the next step and sends the commands to the Crazyflie drones. This enables the control of multiple drones for the purpose of pursuit and capture.

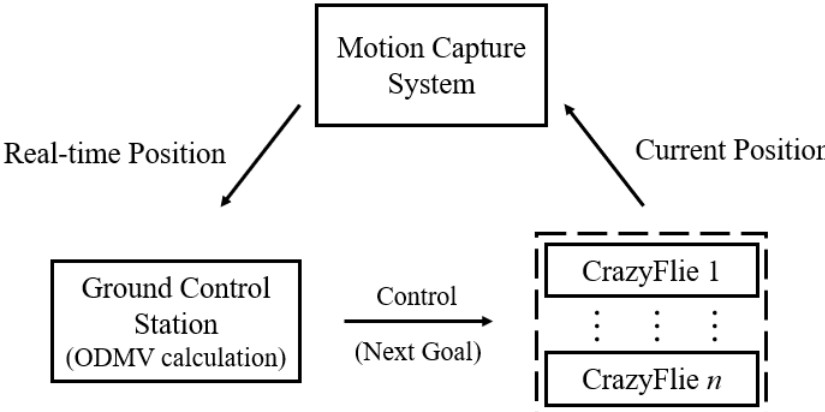

**Figure 17.** Structure of multi-drone capture control system.

The experiment has three different environments: obstacle-free, multiple obstacles, and dynamic targets. In all three cases, the initial positions of the drones are located at the top left, and the target is positioned at the bottom right. We set the parameter of WPA as $d_{max}$ = 5, $d_{min}$ = 0.5. The number of drones used in the experiment is four, and the velocity of the drones is 1.1 m/s. It is worth noting that, in order to avoid crashes caused by the turbulence generated when the distance between drones is too close, we did not minimize the volume of the target 3D-Voronoi cell; the safe distance between each drone is set as 0.3 m. At the beginning of our experiment, we assume that each drone has a good knowledge of the environment, including the position of the target and obstacles.

### 5.1. Obstacle-Free Environment

In the obstacle-free environment, the drones were tasked with capturing a stationary glowing LED. The yellow line represents the connection of the captures. The LED was suspended at the bottom right corner of the experimental scene at a position of (0.9 m, 0.9 m, 1.1 m). The capture process is illustrated in Figure 18.

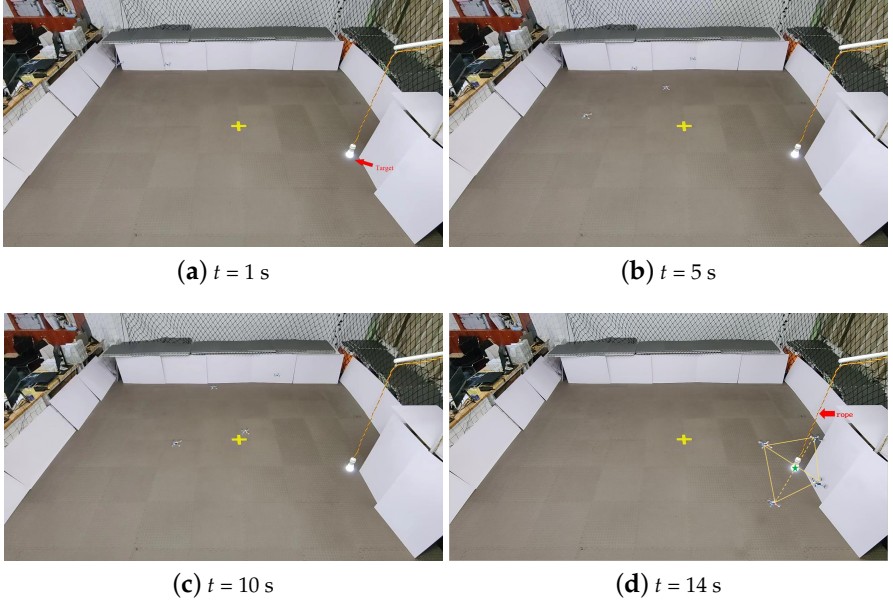

(**a**) *t* = 1 s      (**b**) *t* = 5 s

(**c**) *t* = 10 s      (**d**) *t* = 14 s

**Figure 18.** Multi-drone capture under an obstacle-free environment.

### 5.2. Multiple Obstacles Environment

In the obstacle environment, five cylindrical obstacles were constructed. To avoid interfering with the tracking of the drones by the Optitrack system, the surfaces of the

obstacles were covered with green cloth. The safe distance between drones and obstacles was set as 0.3 m. The capture process in this environment is shown in Figure 19.

From the final distribution in Figure 18d and Figure 19d, it can be observed that the drone formation has effectively captured the target. The final distribution of drones is similar to the configuration shown in Figure 3b for $n = 4$. The yellow line represents the connection of the captures.

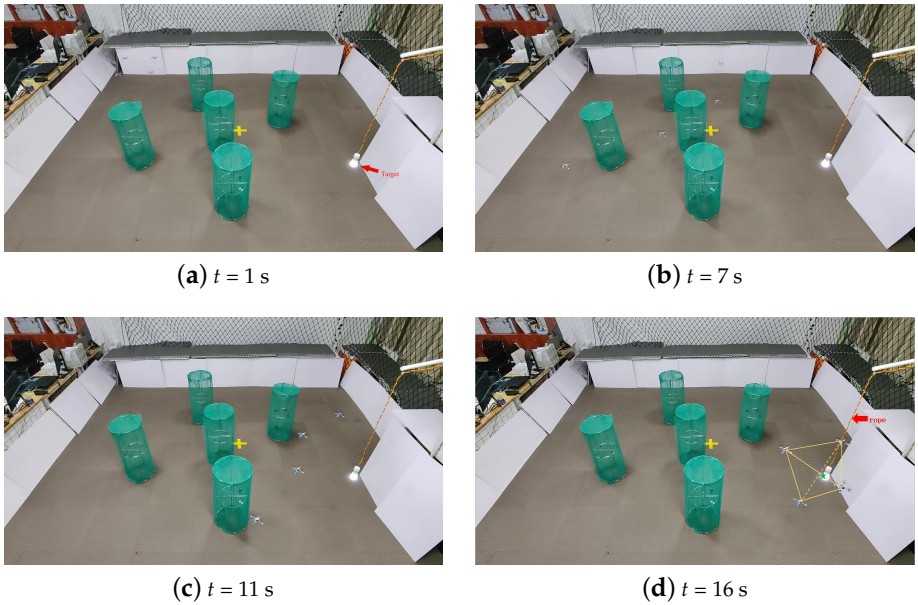

(**a**) $t = 1$ s       (**b**) $t = 7$ s

(**c**) $t = 11$ s       (**d**) $t = 16$ s

**Figure 19.** Multi-drone capture of under obstacles environment.

*5.3. Dynamic Target Environment*

In the dynamic target environment, three drones were designated as the capturing drones, and one drone served as the target. Throughout the entire experiment, the target did not employ any specific evasion strategy. It simply moved linearly from an initial point to another predetermined point. The capture process is illustrated in Figure 20.

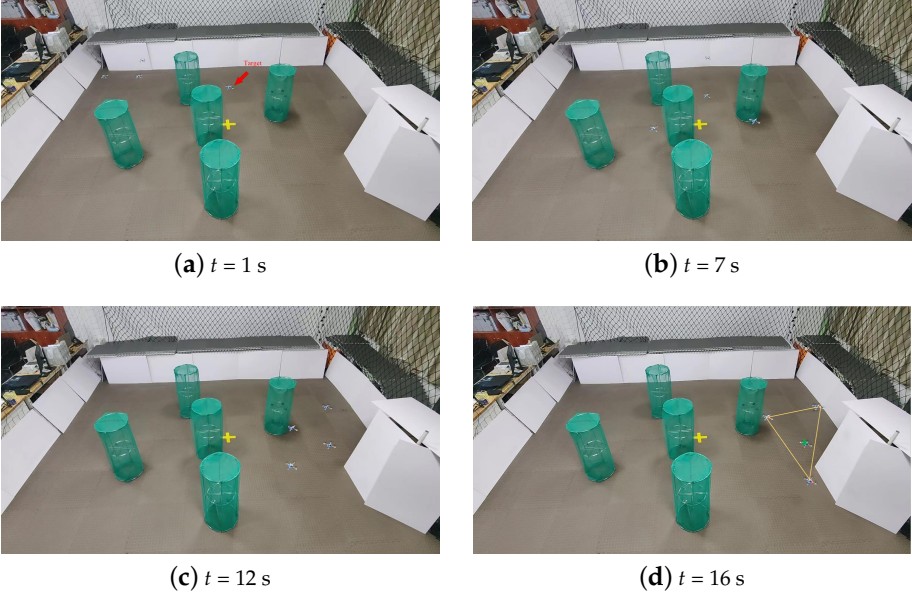

(**a**) $t = 1$ s       (**b**) $t = 7$ s

(**c**) $t = 12$ s       (**d**) $t = 16$ s

**Figure 20.** Multi-drone capture under dynamic target environment.

The final capture result in Figure 20d is similar to the capture effect shown in 3a for $n = 3$.Through the experimental process, it can be observed that under the ODMV strategy, the drone formation is able to capture the target. This conclusion can be extended to capture experiments with a larger number of drones.

## 6. Summary and Outlook

In this paper, 3D-Voronoi is used for the formation strategy of multi-drone systems, and an ODMV strategy is used to achieve the capture of drone formation in a 3D environment. The paper also provides sufficient proof for the reliability of guaranteed capture and omni-directional capture angle. To sum up, the contribution of this article can be summarized as follows:

1.  To generate the capture direction of drone formation in a 3D environment, this paper proposes a capture strategy that combines 3D-Voronoi tessellation with minimum volume strategy. The 3D-Voronoi tessellation can ensure collision avoidance between drones, which can save computational consumption of formation. The minimum volume strategy can provide a capture direction of drones in formation to complete the final capture. The minimum volume capture strategy based on 3D-Voronoi tessellation provides a new way for multi-drone formation to capture the target in a 3D environment. In addition, we have demonstrated guaranteed capture and omni-directional capture angle in a 3D environment.
2.  We have solved the problem of unequal capture angle between drones and target, which will reduce the swing of drones. This paper proposes the ODMV capture strategy, which allows drones to enter the capture formation at a better capture angle. In other words, drones will be distributed more reasonably near the target 3D-Voronoi cell, presenting a better capture effect. Additionally, the ODMV strategy can minimize the volume of the target 3D-Voronoi cell, which will effectively prevent the re-escape of the target. The developed algorithm can keep and form a capture of the target, which means that the algorithm can capture the target and keep it within a polyhedron formed by multiple pursuers.
3.  Based on the above contribution, the wolf pack algorithm (WPA) was introduced as the movement strategy for drones to verify the ODMV. By eliminating the head wolf following mechanism, the direction of drones is changed from head wolf drone to the direction provided by ODMV; this will help drones avoid the local minimum of traditional WPA. We also replaced the fixed step size with a variable one to improve the convergence accuracy of WPA.
4.  The experiment of the 3D-Voronoi ODMV strategy was successfully conducted using physical drones. By utilizing four Crazyflies in conjunction with the motion capture system, the experiments were carried out in complex environments such as with obstacles and dynamic targets (Supplementary Materials).

In the current work, our capture process only considers the scenario of dynamic targets without escape strategies, and the capture process is performed at a constant speed. In future work, we plan to incorporate escape strategies for the target and consider variable-speed capture involving multi-drones. This will enable the capture process to handle more complex scenarios.

**Supplementary Materials:** The following supporting information can be downloaded at: https://www.mdpi.com/article/10.3390/drones7070458/s1, Video S1: Experiment of OMDV capture video.

**Author Contributions:** Conceptualization, Y.-Q.C. and K.C.; methodology, K.C.; software, K.C. and J.Z.; validation, Y.-Q.C. and K.C. and S.G.; formal analysis, K.Y.; investigation, J.Z.; data curation, J.Z. and D.A.; writing—original draft preparation, K.C.; writing—review and editing, K.C., Y.-Q.C. and S.G. All authors have read and agreed to the published version of the manuscript.

**Funding:** This work was supported in part by the funds of the National Natural Science Foundation of China under Grant 62103315, the Key R & D Program Project of Shaanxi Province (No. 2022GY-238, 2022GY-243), and the Key Industrial Innovation Chain Project of Shaanxi Province (No. 2022QFY01-16, 2023-ZDLNY-63).

**Data Availability Statement:** Data sharing is not applied.

**Acknowledgments:** The authors are grateful to the reviewers for their valuable suggestions.And grateful to the Shaanxi Autonomous Systems and Intelligent Control International Joint Research Center for providing the experimental field and MESA Lab.

**Conflicts of Interest:** The authors declare no conflict of interest.

## Abbreviations

The following abbreviations are used in this manuscript:

| | |
|---|---|
| $V$ | 3D-Voronoi cell |
| $q$ | Independent point |
| $p$ | Generators |
| $V_k$ | Target 3D-Voronoi cell |
| $A_k$ | The area of target Voronoi cell in 2D |
| $u_p$ | The motion direction of drones in minimum area strategy |
| $C_i$ | The centroid of adjacent line |
| $x_p$ | Robot position |
| $l$ | The length of adjacent line in 2D |
| $V_{V_k}$ | The volume of target 3D-Voronoi cell |
| $A_j$ | The area of adjacent surface in 3D |
| $x_k$ | Target position |
| $N_n$ | The number of drones |
| $L$ | The distance between any two drones |

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
