# Peer review of "Omni-Directional Capture for Multi-Drone Based on 3D-Voronoi Tessellation"

_drones, doi:10.3390/drones7070458_

Round 1

Reviewer 1 Report (New Reviewer)

This paper addresses the multi-drone formation capture in the 3D environment. The traditional 2D-Voronoi is extended to the 3D environment and the non-overlapping spatial division property of 3D-Voronoi is used to avoid the collision between drones. An omnidirectional area minimum strategy is proposed to complete the effective capture of the target by constraining the capture angle. The wolf pack algorithm (WPA) with variable steps is introduced to provide a movement strategy for multi-drone formations. The simulation and experimental results prove the effectiveness of the proposed method. However, some questions need to be improved and clarified.

(1)  The two sentences in the abstract “When it comes to 3D, most of the existing works on capture have loopholes that will be causing the target still having some direction to escape. Moreover, only considering the capture distance without constraints on the capture angle can also affect the stability of the drones.” this is about the problems that the existing research has not yet solved. So, the above two sentences are suggested to be put in the introduction.

(2)  There is a grammatical error in the sentence in the abstract “When it comes to 3D, most of the existing works on capture have loopholes that will be causing the target still have some direction to escape.” Please check carefully.

(3)  At the end of the paper, the contribution of the article is mentioned. To highlight the significance of the research, it is suggested to briefly write about the contribution and innovation after explaining the problems of other papers in the introduction.

(4)  Figure 4 is not clear, please change it to a vector diagram.

(5)   How to select a proper set of algorithm-related parameters? This point is still not clear. A more detailed study regarding the effect of parameters on the obtained results should be added.

(6)  Please correct the errors in English grammar, spelling, and sentence structure so that the goals and results of the study are clear to the reader.

(7) In recent years, many effective approaches which have the capability to deal with the considered problems have been proposed. For example, some keywords are provided, and the authors can easily find some relevant works on Google Scholar:  "Asynchronous Tracking Control of Leader–Follower Multiagent Systems With Input Uncertainties Over Switching Signed Digraphs"; and "Dynamic Target Enclosing Control Scheme for Multi-Agent Systems via a Signed Graph-Based Approach". It is suggested for the author to compare/comment/discuss the main difference of these newly-developed methods in the paper (maybe in the introduction or the simulation result section). If possible, please also add some comparative simulation results.

Please read through the whole article to optimize the language

Author Response

Dear Reviewer: Please kindly refer to the attached PDF file with replies to your specific comments. We wish this round of revision is now satisfactory. Much appreciated!

YangQuan Chen for coauthors.

Reviewer 2 Report (New Reviewer)

This article presents a new strategy for omnidirectional formation and capture involving multiple agents (UAV) in a 3D environment. For this purpose, the 3D Voronoi method, variable step wolf pack algorithm, and an artificial potential fields algorithm are applied.

In general, the manuscript is clear and well-structured. The theme is not new, but it still has many gaps that can be explored, as demonstrated by the presented work. However, some important points need to be reviewed for a more comprehensive understanding by the readers.

1) How was the decision made to use the artificial potential fields algorithm for obstacle avoidance? As mentioned in the article, you made certain modifications to tackle the problem of unreachable targets. However, what about the issue of the lack of paths and the occurrence of oscillations in the presence of obstacles or narrow passages? Was this aspect evaluated?

2) Additionally, I noticed that there were no tests conducted in the presence of dynamics obstacles. How would the developed algorithm perform in such a scenario?

3) In section 4.1, you refer to figures 5 and 6, but there is no information provided regarding Figure 7, which precisely showcases the result of the algorithm presented by you.

4) In line 148, the variables dmax and dmin are missing from the text.

5) Review the referencing of equations throughout the text. For instance, in line 183, did you mean to refer to Equation 9? The same applies to the figures. In line 336, did you mean to refer to Figure 15?

6)  Something that was not clear in the text: How do the drones know the position of the fixed target and the obstacles in the environment? I believe they are not equipped with a camera or a multiranger for laser sensing. In this case, I assume that the algorithm is provided with pre-stored information about the position of the target and obstacles. At the beginning of the experiment, does the system already have the entire map of the environment? Please clarify this further. In the experiment with the drone as the target, would this information come from the positioning system?

7) The drone formation determines the capture target based on the distance. Please insert this value into the text. Additionally, the capture process is performed at a constant speed. Please include this information about the speed used.

8)  What is the minimum distance between the drones that you evaluated for this project, to avoid turbulence and collision? Please include this information in the text.

9)  Is the algorithm processing done by Matlab on the ground station? From the YouTube video, I noticed that the drone performs a movement and waits for the next information. How much time does it take to send this new position? Have you attempted to embed the entire algorithm processing? The Crazyflie has an expansion that adds processing capabilities. Do you think the algorithm could be embedded and operate in real-time?

Author Response

Dear Reviewer: Please kindly refer to the attached PDF file with replies to your specific comments. We wish this round of revision is now satisfactory. Much appreciated!

YangQuan Chen for coauthors.

Round 2

Reviewer 2 Report (New Reviewer)

Dear authors,

I would like to inform you that all reported observations have been incorporated into the submitted article. You performed a thorough review of the material.

However, when reviewing the second version of the article, I noticed a decrease in the quality of Figures 1 and 2 compared to the previous version. I suggest checking the resolution of the figures.

This manuscript is a resubmission of an earlier submission. The following is a list of the peer review reports and author responses from that submission.

Round 1

Reviewer 1 Report

The authors propose using 3D-Voronoi tessellation as a formation strategy for multi-drone systems and introduce the minimum volume capture strategy. This approach enables multi-drone formations to complete capture tasks in 3D environments successfully. The paper also analyzes and improves the shortcomings of the minimum volume strategy to further enhance the capture process for multi-drone systems.

Section 1. Introduction: This section introduces the context of this work and the related work in this field. However, the introduction needs to be deeper and kept in the scientific context.

Section 2. Collaborative capture strategy based on 3D-Voronoi tessellation: This section describes the shortcomings of previous works that motivated the introduction of a new methodology of 3D-Voronoi capture strategy with omni-directional minimum volume.

Section 3. 3D-Voronoi capture strategy with ODMV: This section applies the minimum area capture strategy to the multi-drone formation capture in 3D space, based on the research in [16]. In this contribution, the authors propose the minimum volume capture strategy in 3D environments and the improved ODMV strategy. However, the proposed strategy is still too generic and lacks a substantial contribution. The proposed strategy assumes that the static target is captured in an obstacle-free environment. It would be more interesting if it were a more complex dynamic environment with moving target and obstacles. So that it can enable the drone formation to adapt to a more realistic complex environment.

In my opinion, some interesting aspects that are missing and can potentially improve the paper as follows:

a. Related to the above, the authors mentioned in Section 3, "Based on the research in [16], this section applies the minimum area capture strategy to the multi-drone formation capture in 3D space. Due to the difference between 3D and 2D space, the minimum area capture strategy becomes the minimum volume capture strategy in 3D space." 

It would be interesting to know how this is beneficial in addition to [16]. If there is a novelty in there, I think this should be stated as a contribution and a more detailed explanation and comparison results should be provided.

b. How does the strategy handle failure cases? Especially in a complex and dynamic environment (obstacle avoidance, distance, speed, etc.,).

Section 4. Simulations and Analysis: In this section, the scenarios are well chosen and motivate the formation of multi-drone strategies. However, the explanations are too shallow, with only simulation results. It would be more valuable to provide real experiments of one or two scenarios, accompanied by figures/videos, rather than to provide Matlab simulation results. Furthermore, the scenarios can be more straightforward, e.g., conducted in a lab or outside environments, similar to Ref [16].

Minor editing of the English language required

Author Response

Dear Reviewr #1: Please kindly refer to the attached replies to reviewer's commend. Thank you for your efforts in helping us to improve our paper.

Reviewer 2 Report

This paper presents a multi-drone 3D capture strategy using an omni-directional minimum volume (ODMV) method. The approach uses a 3D-Voronoi diagram with non-overlapping space division to prevent collisions and introduces a minimum volume strategy and a variable step size wolf pack algorithm (WPA) to effectively capture the target.

The following comments are for the authors to improve the paper and to clarify some concept.

1.   The contributions of this paper should be further explained. A more detailed statement of the motivation for writing this article would have made it more acceptable.

2.   There are many undefined parameters. A table for the notations can be useful.

3.   Provide a figure of logic flow/frame work of your entire work so the readers get a general picture of your work at the first glance.

4.   Add the measurement units labels for abscissa and ordinate for all the figures from the paper.

5.   Add the both the advantages and the disadvantages of the proposed method.

6.   Can you discuss the potential scalability of this approach? How would the approach perform with a larger number of drones or in a more complex 3D environment?

7.   What’s difficulty brought by the new algorithm?

8.   Experiments should be added to strengthen the convincingness of the proposed method.

Author Response

Dear Reviewer #2: Please kindly refer to the attached replies to reviewer's commend. Thank you for your efforts in helping us to improve our paper.

Reviewer 3 Report

1) The abstract is poor. Authors should highlight the context, the gap, the contribution and the relevance of their results.

2) The introduction is like an related work section. The authors must highlight the context of this research. Position the work the context and figure out their contribution and innovation.

3) How can you evaluate our results compared to the current state of the art?

4) How to choose the right initial position of the drones? and what is the impact on the space cover?

5) In what case, the drone can't move outside it voronoi space?

6) In the Figure, you put x1, x2 and x3 instead of x,y,z.

7) What is the impact of each of space parameters (X,Y and Z) on the voronoi space? If, the fix Y,Z and change X what is the impact? and vice-versa for Y, Z.

Finally, this paper need to be more highligted. The presentation is very poor and the experimental results are not concluded.

Need to be improved.

Author Response

Dear Reviewer #3: Please kindly refer to the attached replies to reviewer's commend. Thank you for your efforts in helping us to improve our paper.

Round 2

Reviewer 3 Report

The authors have taken account of all recommendations.

I think that the paper is more improved and can be the subject of publication in this journal.

Thank you for their usefll efforts.